# Gene Targets of CAR-T Cell Therapy for Glioblastoma

**DOI:** 10.3390/cancers15082351

**Published:** 2023-04-18

**Authors:** Chaoqun Wang, Yuntao Li, Lijuan Gu, Ran Chen, Hua Zhu, Xu Zhang, Yonggang Zhang, Shi Feng, Sheng Qiu, Zhihong Jian, Xiaoxing Xiong

**Affiliations:** 1Department of Neurosurgery, Renmin Hospital of Wuhan University, Wuhan 430060, China; 2Department of Neurosurgery, Huzhou Central Hospital, Affiliated Huzhou Hospital, Zhejiang University School of Medicine, Huzhou 310009, China; 3Central Laboratory, Renmin Hospital of Wuhan University, Wuhan 430060, China; 4Huzhou Key Laboratory of Basic Research and Clinical Translation for Neuromodulation, Huzhou 313003, China

**Keywords:** CAR-T, glioblastoma, target, chimeric antigen receptor T-cell immunotherapy

## Abstract

**Simple Summary:**

Glioblastoma is the most prevalent cerebral cancer in adults without proven therapy. Chimeric antigen receptor T cell treatment has demonstrated good clinical success in hematologic cancers. The application of chimeric antigen receptor T cell therapy to solid tumors, such as glioblastoma, is currently the subject of scientific investigation. The creation of chimeric antigen receptor T cells and its iterations are briefly described in this article. Specifically, this study addresses the obstacles and hurdles faced and glioblastoma targets studied in recent years. This review provides the reader with a comprehensive understanding of the current state of chimeric antigen receptor T cell therapy for glioblastoma and a clear understanding of each therapeutic target.

**Abstract:**

Glioblastoma (GBM) is an aggressive primary brain tumor with a poor prognosis following conventional therapeutic interventions. Moreover, the blood–brain barrier (BBB) severely impedes the permeation of chemotherapy drugs, thereby reducing their efficacy. Consequently, it is essential to develop novel GBM treatment methods. A novel kind of pericyte immunotherapy known as chimeric antigen receptor T (CAR-T) cell treatment uses CAR-T cells to target and destroy tumor cells without the aid of the antigen with great specificity and in a manner that is not major histocompatibility complex (MHC)-restricted. It has emerged as one of the most promising therapy techniques with positive clinical outcomes in hematological cancers, particularly leukemia. Due to its efficacy in hematologic cancers, CAR-T cell therapy could potentially treat solid tumors, including GBM. On the other hand, CAR-T cell treatment has not been as therapeutically effective in treating GBM as it has in treating other hematologic malignancies. CAR-T cell treatments for GBM have several challenges. This paper reviewed the use of CAR-T cell therapy in hematologic tumors and the selection of targets, difficulties, and challenges in GBM.

## 1. Introduction

Glioma is the most prevalent primary malignant brain tumor, accounting for approximately 81% of all central nervous system (CNS) malignancies [1]. Gliomas are divided into four categories in the fifth edition of the World Health Organization Classification of Neoplasms of the Central Nervous System (2021): adult-type diffuse glioma, pediatric-type diffuse low-grade glioma, pediatric-type diffuse high-grade glioma, and restricted astrocytic glioma [2]. Glioblastoma (GBM) is a grade IV intracranial tumor with astrocytic differentiation [3]. GBM accounts for 12–15% of all cerebral tumors and has the greatest malignant degree and the poorest prognosis among gliomas, with a 5-year survival rate of just 5% [4]. Most GBMs are naturally infiltrative and develop in diverse regions of the supratentorial cerebral hemispheres, frequently infiltrating several lobes and deeper tissues. The treatment strategies for GBM are surgery, radiation, tumor electric field therapy, and immunotherapy. The current standard of care for GBM comprises temozolomide (TMZ) chemotherapy, fractionated external irradiation as postoperative adjuvant radiation therapy (RT), and surgical excision of as much of the tumor as possible without worsening neurological damage [5]. Despite the combination of treatment modalities, the patients of GBM have a short survival cycle, with overall survival (OS) of approximately 20 months post-diagnosis [6,7] The tumor heterogeneity of GBM, the poor demarcation between GBM tumor cells and normal brain tissue, and the challenge of entirely separating the lesion following surgery are the major causes of the poor prognosis [8]. Poor prognosis is also associated with the blood–brain barrier (BBB), which keeps the brain relatively immune [9].

CAR-T cell immunotherapy, generally known as CAR-T therapy, is a novel immunotherapy against cancer [10]. This method extracts T lymphocytes from the patient’s blood and genetically modifies them in vitro through genetic modification techniques, causing the T cells to express specific antigen recognition structural domains, and then the T lymphocytes are injected back into the patient to treat disease [11] (Figure 1).

CAR-T cells effectively treat blood-related tumors, particularly in relapsed, refractory B-cell-derived malignant hematologic tumors [12]. Two children with acute lymphoblastic leukemia who received the first CAR-T therapy in 2013 achieved a complete response (CR) [13]. In 2020, the Food and Drug Administration (FDA) approved two CAR-T cell therapies for treating CD19+ B cell cancers [14]. Inspired by the successful CAR-T cell therapy in hematologic tumors, CAR-T cell therapy has been applied in the treatment of GBM.

CAR-T cells are prepared in six steps. The patient’s peripheral blood is collected (①). Peripheral blood mononuclear cells are isolated by centrifugation (②). T cells are then enriched based on amoetoid isolation and immunomagnetic beads (③). The CAR gene is introduced into T cells using a vector (④). Activation of CAR-T cells in vitro and expansion of CAR-T cells to the therapeutic dose is required. However, composition, purity, safety, and efficacy must be determined before being used in clinical trials (⑤). Finally, the patients are treated with CAR-T cells (⑥).

## 2. CAR-T Cells

Adoptive cell transfer therapy (ACT) refers to using self-generated immune cells against tumor cells, modified and expanded by various means, followed by their reintroduction into the body to stimulate an immune response to kill tumor cells [15]. ACT has greatly progressed in leukemia, melanoma, and other solid tumors [16]. CAR-T therapy operates on the central principle of using the patient’s immune cells to eliminate diseased cells [17]. Presently, the fundamental structure of most CAR molecules consists of four components, including an extracellular structural domain consisting of a single-chain variable fragment (scFv), a transmembrane structural domain, a hinge region, and an intracellular structural domain (primarily CD3ζ) [18].

CARs can be broadly categorized into four generations based on their intracellular structural domains: the first generation of CAR-T cells lacks any co-stimulatory signals; the second generation has a single co-stimulatory signal; the third generation has two or more co-stimulatory signals, and the fourth generation consists of precision CAR-T cells. SvFv and the intracellular structural domain (CD3) comprised the first generation of CAR molecules. The first generation of CAR molecules has a reasonably simple design without required co-stimulatory signals, demonstrating that CAR-T cells cannot efficiently proliferate, release cytokines, or show significant antitumor activity [19]. The intracellular structural domain of the second generation of CAR molecules introduces a co-stimulatory molecule (4-1BB and CD28) based on the first generation’s CAR molecule. This enhances the expansion and tumor-killing effect of CAR-T cells in vivo due to the additional co-stimulatory signals obtained by T cells [20,21]. The third-generation CAR has two or more co-stimulatory molecules (OX40 and DAP10). Some preclinical studies have found that 4-1BB combined with CD28 co-stimulatory factor can enhance CAR-T cell proliferation and prolong its activity [22]. However, the third-generation CAR-T cells have been shown to cause a decrease in T-cell stimulation threshold, leading to a large amount of cytokine release and affecting the tumor-killing activity of third-generation CAR-T cells [23,24]. Current research is unable to determine whether the second or third generation is superior. The CAR-T cells of the fourth generation, T-cells redirected for universal cytokine killing (TRUCK), are genetically altered to secrete particular cytokines, such as interleukin (IL)-12 and express co-stimulatory ligands, and attract and activate other immune cells for an immune response to enhance the antitumor activity of CAR-T [25]. Currently, the fifth generation of CAR (universal CAR) uses gene editing technology to remove certain genes to prevent rejection and lengthen the survival period of CAR-T cells within the body, improving their capacity to eradicate malignancies [26]. Today’s most widely used CAR-T technology is the second generation, which is more developed and supported by a large body of clinical trial data (Figure 2).

## 3. CAR-T Cell Therapy in Hematological Tumors

With the unremitting efforts of researchers, CAR-T cell immunotherapy has progressed in treating a wide range of hematological malignancies. A prevalent hematologic malignancy is acute B-cell lymphoblastic leukemia (B-ALL). Despite chemotherapy or hematopoietic stem cell (HSC) transplantation, the prognosis for some patients with refractory or relapsed (r/r) B-ALL is less than optimal. The treatment of B-ALL and r/r B-ALL has benefited significantly from CAR-T, a potential tumor immunotherapy [27,28,29,30,31].

Nevertheless, the use of CAR-T to treat acute T-lymphoblastic leukemia (T-ALL) is difficult since both healthy and malignant T cells produce many target antigens [32]. Yet many studies have reported that CAR-T in T-ALL patients successfully induced antitumor effects and sustained proliferation [33,34,35,36,37,38].

CAR-T therapy for refractory/relapsed multiple myeloma (RRMM) has demonstrated promising clinical potential [39,40,41]. B cell maturation antigen (BCMA), one of the several targets of CAR-T treatment for myeloma, is expressed on roughly all MM cells. Thus, BCMA is a good and secure target for MM [42].

## 4. CAR-T Cell Therapy in GBM

CAR-T cell therapies are effective in several hematologic malignancies. CAR-T cell treatments targeting solid tumor antigens such as GBM have been the subject of many clinical studies (Table 1). Currently, preclinical and clinical studies of CAR-T therapy have been conducted on all solid tumors. In the phase I clinical trial of gastric cancer patients, the objective response rate (ORR) and disease control rate (DCR) of the 37 enrolled patients were 48.6% and 73.0%, respectively, with a median progression-free survival (mPFS) of 3.7 months. The 6-month duration of response (DOR) was 44.8%, and the 6-month OS was 80.1%. Phase II clinical trials are currently underway [43]. In 2022, results from a recent phase I clinical trial (NCT03089203) of a targeted prostate-specific membrane antigen (PSMA) CAR-T therapy for metastatic castration-resistant prostate cancer (mCRPC) were published in Nature Medicine [44]. The study included 18 patients, and 1 patient displayed a large number of clones of CAR-T cells and a decrease in prostate antigen (PSA) of more than 98% following the treatment. Moreover, 4 patients had a more than 30% decrease in PSA. However, cytokine release syndrome (CRS) as the side effect of CAR-T was also observed in five patients. A phase I clinical trial was conducted in patients with glypican-3 (GPC3) and mesothelin (MSLN)-positive advanced hepatocellular carcinoma (HCC), pancreatic cancer (PC), and ovarian cancer (OC) [45]. One HCC patient had complete disappearance of the lesion after day 32 of the intratumoral injection of CAR-T. One patient with PC had almost complete disappearance of the tumor at 240 days of treatment. In this clinical study, investigators designed CAR molecules that can secrete IL-7 and CCL19 to enhance T-cell infiltration and CAR-T cell survival.

While many more clinical trials demonstrate promising results, CAR-T still has significant challenges and difficulties for solid tumors. This might be due to several factors, including the distinct tumor microenvironment (TME) of the nervous system, the absence of particular targets, the maintenance of homing and activation, tumor heterogeneity, antigen loss, on-target off-tumor effects, and security of CAR-T applications [46]. Nevertheless, recent preclinical investigations exposed that CAR-T treatment for GBM is feasible and secure [47]. Yet numerous obstacles and challenges are associated with CAR-T therapy for GBM.

Bibliometrics is a subject that visualizes dynamic features such as subject hotspots and development trends by using mathematical and statistical methods to analyze the structure and interrelationship of literature information quantitatively [48]. By analyzing the literature in a particular field, researchers can better understand research trends, hot spots, and significant articles and authors more accurately. We conducted a bibliometric analysis of the keywords CAR-T cells and glioblastoma to obtain the countries or regions with the most studies in this field. By analyzing the number of national publications, one can roughly determine the important countries in the particular field, which is convenient for researchers to focus on the relevant countries in future research work.

We searched the Web of Science Core Collection using the search formula: TS = ‘Chimeric Antigen Receptor Therapy’ OR ‘CAR T-Cell Therapy’ OR ‘CAR T Cell Therapy’ OR ‘T-Cell Therapies, CAR’ OR ‘T-Cell Therapy, CAR’ OR ‘Therapies, CAR T-Cell’ OR ‘CAR T-Cell Therapies’ OR ‘Therapy, CAR T-Cell’ AND ‘Glioblastomas’ OR ‘GBM’ OR ‘Grade IV Astrocytoma’ OR ‘Grade IV Astrocytomas’ OR ‘Glioblastoma Multiforme’. The document type was only the article, and there were no other restrictions. Visualizing the number of publications by country, we found that the United States (U.S.) (106) leads the world in CAR-T and GBM research and is ahead of China (38), which ranks second (Figure 3).

### 4.1. Potential Targets of CAR-T Cell Therapy for GBM

Due to the highly heterogeneous nature of GBM, it is challenging to screen for adequate tumor surface antigens [49]. Furthermore, tumor surface antigens can be classified according to their specificity: tumor-specific antigen (TSA) and TAA [50]. In GBM, there is a relative lack of TSA [51]. CAR-T targets mostly TAA, often expressed in small amounts in normal tissues [52]. Consequently, CAR-T cells invariably harm normal tissues expressing the target antigens while simultaneously eradicating tumor cells. The receptor for human epidermal growth factor type 2 (HER2), IL-13 receptor alpha 2 (IL-13Rα2), CD133, B7-H3 (type I transmembrane protein), ganglioside (GD2), and human epidermal growth factor receptor variant III (EGFR vIII) are the principal targets exploited in GBM currently. Recent research indicates that most GBMs express the Advillin (AVIL) antigens, whereas AVIL is expressed slightly in normal tissues. Hence, AVIL is anticipated to be a novel target for CAR-T cell treatment [53]. In the following section, the main targets are briefly described (Figure 4).

#### 4.1.1. EGFR vIII

Epidermal growth factor receptor variant III (EGFRvIII) is the most common epidermal growth factor receptor, which is a product of 801 bp rearrangement of deletion in the coding sequence box of the EGFR extracellular domain, resulting in deletion of residues 6 to 273 and insertion of glycine as residue 6 [54]. EGFR vIII is a typical tumor-specific antigen that is highly expressed on the cell surface of GBM and other tumors, while it is low or absent in normal tissues [55,56,57,58]. In addition to being a therapeutic target for CAR-T cells, EGFR vIII can also be used as a biochemical indicator of the prognosis of glioblastoma patients after surgical treatment [59]. EGFR vIII was first identified in primary human GBM, with an expression rate of approximately 30% [60]. In 2014, Hongsheng Miao observed that EGFR vIII-CAR-T cells proliferated in the tumor region and inhibited the growth of tumor cells, improving the survival rate of experimental mice [61]. In 2017, Donald M. O’Rourke et al. conducted the first clinical study of human intravenous administration of a single dose of autologous CAR-T cells for the treatment of 10 cases of recurrent glioblastoma (GBM) by retargeting the CAR molecule to EGFR vIII [62]. Of these, seven patients underwent post-surgical intervention with CART- EGFR vIII and were found to have metastasized CART-EGFR vIII cells to tumor cell areas. In five of these seven patients, antigen reduction was observed. Abbott et al. constructed GCTO2 (a novel scFv that recognizes EGFRvIII) CAR-T and evaluated the function of GCTO2 CAR-T using the U87- EGFR vIII glioblastoma model. The regression of tumor cells was observed after a single intravenous injection of GCTO2 CAR-T cells, which is highly specific for EGFR vIII. However, the affinity was much lower than Clone 2173 (C2173) scFv [63].

Due to the presence of TME, CAR-T cells targeting only EGFR vIII do not achieve durable antitumor activity. However, when a single dose of IL-12 is injected locally, with a smaller systemic immune response, IL-12 can act as a potent adjuvant capable of remodeling TME and enhancing the cytotoxicity of CAR-T cells. Moreover, IL-12 can increase the infiltration of proinflammatory CD4+ T cells and decrease the amount of regulatory cells (Tregs) to produce long-lasting antitumor action [64].

Programmed death receptor 1 (PD-1) is the most common immune checkpoint found in T cells [65]. PD-1 has been linked to numerous tumor cell alterations, the tumor immune microenvironment, and T lymphocyte infiltration [66]. Furthermore, researchers created universal CAR-T cells resistant to PD-1 inhibition by using the CRIPSR-Cas9 system to investigate whether inhibition of PD-1 enhances the tumor-killing activity of CAR-T cells in preclinical models of human GBM [67]. In vitro, the investigators discovered that CAR-T-EGFRvIIIΔPD-1 cells generated more Th1 proinflammatory cytokines than CAR-T-EGFRvIII cells, sustaining CAR-T cell proliferation for longer. In addition, compared with controls, CAR-T-EGFRvIIIΔPD-1 cell injections into the caudal vein and ventricle dramatically increased the lifespan of human GBM mice. A phase I study of the PD-1 inhibitor pembrolizumab in combination with CAR-T in patients with MGMT (O6 -methylguanine-DNA methyltransferase)-unmethylated GBM (NCT03726515) is also underway.

#### 4.1.2. HER-2

The EGFR family member HER-2, also known as ErbB2, is overexpressed in brain tumor cells. It is found on human chromosome 17 [68]. The protein is also involved in the development of many tumors, such as gastric cancer, OC, and osteosarcoma. HER-2 first appeared in breast cancer studies. The high HER-2 expression is associated with poor prognosis and survival in patients.

The first patient passed following the therapy with HER2-targeted CAR-T, raising safety concerns about the molecular targeting of HER-2. The patient received lymph-depleting chemotherapy prior to intravenous infusion of 10^10 HER-2 targeted CAR-T cells. It is hypothesized that death was because of the tumor off-targeting of normal lung epithelial tissue, resulting in massive cytokine release and triggering respiratory distress and pulmonary edema. Studies found that HER-2-specific T cells generated by GBM patients have potent antitumor activity against their own HER-2-positive tumor cells, including CD133-positive GBM stem cells [69]. In a clinical trial, 8 patients had illness progression, 7 patients were stable for eight weeks to 29 months, and 1 patient had a partial response for 9 months [70].

Currently, three clinical studies are testing intracranial injection of anti-HER-2 CAR-T (NCT01109095, NCT02442297, and NCT03500991) for treating CNS malignancies, including GBM. Recurrent intracranial injection of HER-2 CAR-T on a cohort of pediatric and adolescent patients with CNS malignancies was shown to be safe, well tolerated, and capable of generating an immune response according to early results from the phase I research “BrainChild-01” [71]. Young adults and children with diffuse midline glioma and other resistant or recurrent CNS malignancies participated in a phase I clinical study (NCT03500991). Repeated local delivery of HER2 CAR-T cells led to an increase in the emission of chemokines such as CCL2 and CXCL10 in cerebrospinal fluid (CSF) without any obvious dose-related toxicity of the CAR-T [71].

#### 4.1.3. IL-13Rα2

IL-13Rα2 is the most widely studied target [72]. W. Debinski et al. found that the receptor for IL-13 is abundantly expressed in human GBM, while there are almost no IL-13 binding sites in the normal human brain, and this selective expression makes it an ideal target for GBM in CAR-T cell therapy [73]. Moreover, with the increase in the GBM malignancy, the expression of IL-13Rα2 also increases; thus, IL-13Rα2 is considered to be a prognostic indicator of GBM. Christine E. Brown reported an increase in the number of cytokines and immune cells in the CSF and the regression of tumor cells in the spinal canal and spine after studying a patient with recurrent multifocal GBM treated with IL-13Rα2- CAR-T [47].

Darya Alizadeh found that CAR-T makes a lot of sense as an adjuvant [74]. It can destroy tumor cells directly and stimulate the host’s innate immune system and cytotoxic T lymphocytes to kill tumor cells. Moreover, it was shown that IL-13Rα2 CAR-T increases the effectiveness of CAR-T cells by influencing FN-γ signaling to activate host immune cells.

The scFv employed in CAR-T techniques is usually derived from murine sequences, an important factor in generating anti-CAR immune responses [75]. Nevertheless, human anti-mouse immune responses are triggered by scFv of monoclonal antibodies made from mice, restricting CAR-T’s effectiveness in eradicating tumor cells [76]. It has been found that in vitro and xenograft mouse models transfected with human-derived anti-IL-13Rα2 CAR and murine anti-IL-13Rα2 CAR human-derived CAR-T cells have better amplification ability and more effectively inhibit the growth of GBM tumor cells [77]. Furthermore, Brown et al. created off-the-shelf CAR-T cells that could target IL-13R2 in GBM patients using the immunosuppressive effects of the glucocorticoid dexamethasone. They also conducted the first human evaluation of a glucocorticoid-resistant allogeneic CAR-T cell product for treating recurrent GBM [78].

#### 4.1.4. CD70

CD70 belongs to the tumor necrosis factor superfamily (TNFSF) of transmembrane proteins, while CD27 is its receptor [79]. CD70 and CD27 could modulate the immune response by enhancing T and B cell activation, proliferation, and differentiation. CD70 is mainly expressed in activated T cells, B cells, and mature dendritic cells and is overexpressed in gliomas, with low expression in most normal tissues [80]. CD70 directly participates in immunosuppression by mediating the production of pretumor chemokines such as IL-8, specifically inducing CD8+T cell death [81]. In 2022, a systematic study evaluating the effectiveness of CD70-targeted CAR-T cells in a GBM mouse model demonstrated that CD70 plays a crucial role in the sphere formation and value addition of GBM [82]. A recent preclinical study illustrated that CD70 targeted CAR-T by combining with oHSV-1 in GBM treatment, which increased the ratio of T cells to natural killer (NK) cells in TME, reducing regulatory T cells, promoting IFN-γ release, and ultimately enhancing the therapeutic effect of CD70-specific CAR-T cells [83].

A phase I clinical trial in CD70-positive, MGMT-unmethylated adult GBM was published in 2022 (NCT05353530). Patients underwent intravenous administration of CD70 CAR-T modified by IL-8 receptor (CXCR2) cells (8R-70CAR-T) following maximal tumor reduction surgery and standard radiotherapy. Safety and feasibility were evaluated at day 28 and week 10 after intravenous administration of CAR-T cells, respectively.

#### 4.1.5. B7-H3

B7-H3 (CD276) is a type I transmembrane protein from the B7 immunological co-stimulatory and co-inhibitory family. In humans, chromosome 15q24 encodes B7-H3, consisting of an extracellular region, a transmembrane region, and a short intracellular region. Immunohistochemical (IHC) analysis showed that B7-H3 is only lowly expressed in a few tissues. It is overexpressed in various solid tumors, including GBM [84]. Additionally, in a clinical trial, a 56-year-old patient with recurrent GBM in the left frontal and parietal lobes was treated with targeted B7-H3 CAR-T cells. The researchers evaluated the patient’s clinical results. Seven rounds of CAR-T cell administration were carried out. According to the magnetic resonance imaging (MRI) results, recurrent tumors had significantly decreased after the first round of infusion; however, in cycles six and seven, the patient experienced altered consciousness and drowsiness with visible tumor recurrence [85]. Although this trial did not achieve the expected clinical outcome, it supports the potential of B7-H3-targeted CAR-T for GBM treatment.

Recent studies have shown that CAR-T cells targeting B7-H3 do not achieve satisfactory tumor-killing efficacy. Nonetheless, by building an oncolytic adenovirus (oAds) with a chemokine CXCL11, promising results were observed with CAR-T therapy in a GBM mouse model. CXCL11 armed with oAds increased CAR-T cell invasion of GBM tumors. In addition, oAds selectively kill tumor cells and enhance the infiltration of NK cells, M1 macrophages, and CD8 T lymphocytes through metabolic reprogramming of the immunosuppressed TME [86]. In 2022, there were two phase I clinical trials of targeted B7-H3-CAR-T therapy in GBM patients (NCT05366179 and NCT05241392).

#### 4.1.6. EphA2

Adrenoceptors (Eph) are a vital receptor tyrosine kinase. Currently, there are 14 Eph receptors and 8 related ligands [87]. EphA2, a member of the Eph receptor family, is expressed at a high level in tumor tissues and relatively at a low level in most normal adult tissues. A recent study shows that EpHA2 can act as a potential receptor for platelet-derived growth factor subunit A (PDGFA), promoting PDGFA signaling in combination with platelet-derived growth factor receptor alpha (PDGFRA) and mediating resistance of GBM cells to PDGFRA inhibitors [88]. Multiple independent groups have developed different EPHA2-specific CAR-T cells for treating GBM with encouraging preclinical results. A recent study used GeneArt technology to synthesize two different sequences encoding EphA2-scFvs (D2-1A7 and D2-1B1), yielding two molecules targeting different epitopes of EphA2. Combining two different epitopes of EphA2 and third-generation CAR-T molecules to target GBM humanized mouse models. Based on the outcomes, EphA2 CAR-T cell therapy greatly improved the survival rate of the GBM humanized mouse model, and the antitumor efficacy of EphA2 CAR-T cells was connected to CXCR-1 / 2 overexpression and appropriate IFN-γ production. By triggering the overexpression of PD-L1 in GBM cells, CAR-T cells with elevated IFN-expression demonstrated poor anti-tumor effectiveness. Yet when PD-1 blockers were administered to CAR-T with strong IFN- expression, the anti-tumor activity of the CAR-T cells enhanced once again [89].

Lin, Qingtang et al. reported the first human trial using targeted EphA2-CAR-T to treat patients with recurrent GBM. They observed 3 patients receiving the starting dose of CAR-T (1 × 10^6^ cells/kg) and found that 2 patients developed grade 2 CRS with pulmonary edema, 1 achieved stable disease, and 2 achieved disease progression with an OS ranging from 86–181 days. The grade 2 CRS disappeared in the patients after treatment with dexamethasone [90].

#### 4.1.7. CD133

CD133 antigen with five transmembrane regions is a unique marker of HSCs. CD133 antigen is recognized by three CD133 antibodies: clone AC133, 293C3, and AC141. CD133 has the biological properties of tumor stem cells: autonomous replication, autonomous differentiation, multiplication, and a highly tumorigenic capacity for the organism. Previous studies have found that CD133 marks self-renewing cancer stem cells (CSC) in various solid tumors. Recently, Sheila Singh, a researcher from Canada, developed an immunotherapy for GBM targeting CD133 [91]. Singh et al. designed three immunotherapies, including CAR-T therapy, based on human anti-CD133 antibody fragments targeting unique epitopes present in glycosylated and non-glycosylated CD133, and investigated their targeting of CD133+ cells in a GBM humanized mouse model. The results showed that all three approaches exhibited activity against CD133+ GBM tumor cells, but CD133- CAR-T was the most effective in the GBM mouse model. The researchers also conducted safety studies in a humanized mouse model to explore the potential effects of CAR-T therapy on hematopoietic function. The results demonstrated that CD133-specific CAR-T therapy does not adversely affect normal CD133 HSCs and does not induce acute systemic toxic reactions in the in situ GBM humanized mouse model.

IKZF3 is a member of the IKZF family of proteins, which can mediate cytokine signaling and the growth of immune cells [92]. Through the IKZF3 gene knockout, the activation and added value of T cells and cytokine signaling became more active, and the anti-GBM activity by CD133-specific CAR-T was also improved [93]. Similarly, BAY 60-6583, an adenosine A2b receptor agonist, enhanced the activity of CAR molecules [94].

However, some studies have pointed out that CD133, a glioma stem cell surface marker, has certain limitations. Joo et al. found that both CD133-positive and -negative cells obtained from GBM patients could produce GBM tumor chunks in the brains of immunodeficient mice, while brain tumor chunks containing CD133-negative cells showed higher proliferative and angiogenic ability than brain tumor chunks generated from CD133-positive cells [95]. This result contradicts what has been discussed above, but the effectiveness of any antigen is subjected to repeated trials. Further studies are needed on whether CD133 can be used as an effective target antigen for GBM and how to improve the efficacy of CAR-T.

#### 4.1.8. GD2

Gangliosides are cell surface glycolipid antigens consisting of ceramide, N-acetylneuraminic acid (Neu5Ac), and a nine-carbon skeleton of sialic acid. Gangliosides comprise 6% of all phospholipids and are located on the cell surface and mainly in the nervous system. Gangliosides in the nervous system are GD2. GD2 is uniformly expressed in neuroblastoma and most melanomas and to varying degrees in various other tumors. GD2-CAR-T cells are currently the most efficient in solid tumors, and it is anticipated that they will be the first solid tumor CAR-T cell target to enter phase III clinical trials. In the phase I clinical trial (NCT04196413), GD2-targeted CAR-T cell treatment was evaluated in diffuse midline glioma, with three of four patients showing substantial improvement in neurological impairments and radiological improvement. In addition, no targeted non-tumor toxicity was observed. To enhance the tumor-killing effect, some investigators have used in vivo microscopic imaging to assess the synergistic effect of RT on CAR-T cell therapy. Treatment of GBM mice using GD2-CAR combined with scFv from 14g2a antibody with the murine CD28 and CD3-ζ signaling structural domains on the MSGV1 retroviral backbone revealed RT-induced CAR-T aggregation and extravasation within the tumor mesenchyme within 24 h after intravenous cell delivery and promotion of rapid CAR-T proliferation within 5 days. Moreover, RT allowed rapid extravasation of CAR-T cells from the vascular system and expansion in the TME, resulting in a more robust and durable immune response [96]. In addition to GBM, clinical trials for CAR-T therapy in other gliomas are actively underway. The results from the phase I clinical study of four patients with diffuse endophytic glioma and midline glioma of the spinal cord, which were given GD2 CAR-T cells, have been published [97]. High levels of proinflammatory cytokines in plasma and CSF were associated with clinical and imaging improvements in 75% of patients without any on-target or extra-tumor injury.

#### 4.1.9. Chlorotoxin (CLTX)

Chlorotoxin, a peptide isolated from the venom of the giant Israeli yellow scorpion, is a defensive component secreted by the scorpion venom gland. The main components of CLTX are proteins and non-proteins, among which proteins mainly consist of 20–80aa of various toxic small peptides (2–9 kDa), selectively binding to the potassium and sodium ion channels and calcium on animal cell membranes and affecting the state of ion channels. In GBM, CLTX inhibited the invasive growth and metastasis of glioma cells by specifically binding to glioma cell chloride channels and matrix metalloproteinase 2 (MMP2). CLTX is highly expressed in glioma cells but not in normal brain tissue. It has two kinds of activity, tumor binding and antiangiogenic, respectively. In 2020, researchers developed and tested the first chlorotoxin-targeted T-cell-based CAR-T therapy to target tumor cells in GBM patients in conjunction with the first human clinical trial using chlorotoxin CAR-T cell therapy. It was found that CLTX-CAR-T cells recognized and killed a large number of GBM tumor cells. In addition, the researchers found that CLTX-CAR-T cells largely avoided damaging non-tumor cells in the brain and other organs. In both in vitro cellular and animal experiments, CLTX has been shown to kill human GBM with a high degree of selectivity and specificity in the absence of side effects such as almost cytotoxic and off-target effects. Notably, CAR-T targeting CLTX requires the expression of MMP2 on tumor cells to bind and kill tumor cells effectively [98].

Two phase I clinical studies in MMP2-positive recurrent or progressive GBM (NCT05627323 and NCT04214392) were conducted in 2019 and 2022, respectively, and neither result has been published. Both clinical studies involve the infusion of CAR-T cells targeting CLTX into patients via a dual delivery (intracavitary/intratumoral (ICT) and intraventricular (ICV)) approach to investigate the safety and optimal dosing.

#### 4.1.10. NKG2D

NKG2D is an active receptor found on the surface of T cells, and its gene is located in the NK gene complex. It is involved in innate immunity, facilitating NK cells to recognize virus-infected cells and destroy tumor cells [99]. It is a type II transmembrane receptor with a sequence similar to C-type lectin [100]. Most cancer types, including multiple myeloma, OC, and lymphoma, have been shown to respond favorably to NKG2D-expressing CAR [101]. Yang Dong et al. reported high expression of NKG2DLs (a ligand of NKG2D) in human GBM by flow cytometry and IHC. In addition, by constructing NKG2D-CAR-T cells, it was found that NKG2DCAR-T cells effectively lysed GBM tumor cells and produced high levels of cytokines, perforin, and granulzyme B [102].

The co-expression of IL12 and IFNα2 with CAR-T promoted the proinflammatory TME, reduced T cell depletion, and increased the anti-glioma activity of CAR-T compared with NKG2D expressing CAR-T alone [103]. To assess the security and effectiveness of NKG2D CAR-T cell treatment in patients with recurrent and resistant GBM, a pilot phase I study (NCT04717999) was started in 2021 and is anticipated to be completed in December 2023.

#### 4.1.11. Some Promising Targets

A transmembrane protease called carbonic anhydrase 9 (CAIX), triggered by an oxygen-rich environment, is abundantly expressed in the membranes of most solid tumor cells due to hypoxia [104]. The upregulation of CAIX in GBM has been linked to poor patient prognosis and short patient survival. CAIX controls GBM motility and monocyte adherence in hypoxic environments via EGFR/STAT3 [105]. The effectiveness of CAIX CAR-T cells was assessed by Cui J et al. in a xenograft mouse GBM model made from U251 cells expressing luciferase. A 20% probability of tumor recurrence was reported after two months, and bevacizumab was shown to boost CAR-T effectiveness [106]. The performance of CAIX CAR-T cells was improved by LB-100, a water-soluble protein phosphatase 2A (PP2A) inhibitor [107].

Most metastatic prostate tumors overexpress PSMA, a non-secretory membrane enzyme, while normal tissues only physiologically express PSMA at low levels [108]. Currently, the world’s first phase I trial of CAR-T therapy in prostate cancer demonstrated that CAR-T-PSMA-TGFβRDN has good antitumor activity [44]. In addition, it has been found that PSMA is highly expressed in the vascular endothelium of GBM tissue, and PSMA can promote GBM angiogenesis by interacting with integrin β4 (ITGB4) and activating the NF-κB signaling pathway [109]. However, until now, there have been no studies of PSMA-targeting CAR-T therapy in GBM. Therefore, we believe that PSMA is a target with great research potential.

### 4.2. Difficulties and Challenges

Glioblastoma has a high degree of tumor heterogeneity, which leaves no single antigen that can be used as a universal target covering the entire tumor [110]. As the disease progresses, tumor cells differ in growth rate, aggressiveness, and drug sensitivity, which is the root cause of drug resistance in glioblastoma [111]. In addition, for a single target, there may be target downregulation or loss, resulting in antigen escape. In response to this phenomenon, researchers have designed multi-target CAR-T, which has been proved to be more efficient in identifying and killing tumors. For example, the design of dual-directed CAR-T that can target CD19 and CD22 can effectively solve the disadvantage of single-target resistance theoretically, which has been proved clinically. Trivalent CAR-T has been investigated, for example, for GBM patients, and the development of trivalent CAR-T cells (targeting both HER2, IL-13Rα2 and EphA2) can overcome antigenic heterogeneity and improve treatment outcomes. In addition to searching for TSA, modification of CAR-T cells can also improve their recognition of tumor antigens. WendellLim et al. inserted the synNotch system into CAR-T. Under the control of synNotch, the TAA-related CAR was only produced on T cells that traveled to the tumor and did not assault cells in normal tissues, considerably enhancing the specificity of CAR-T recognition antigens [112]. CAR-T cells transmit RN7SL1, an RNA derived from the body’s own cells, to the tumor through extracellular vesicles (EVs), which stimulate the body’s own T cells to seek cancer cells that do not recognize CAR-T cells. Moreover, RN7SL1 enhanced CAR-T cell activity by encouraging CAR-T cell proliferation and effector memory phenotype development. CAR-T cells that expressed RN7SL1 were found to have better tumor infiltration, longer half-lives, and stronger antitumor activity [113,114].

Facing the characteristic of tumor heterogeneity is another challenge, as heterogeneity is a double-edged sword. It can be used to provide individualized treatment for patients according to heterogeneity. For now, tumor heterogeneity is more of a predisposing factor for the failure of various therapeutic approaches. Increasing the number of antigens covered by CAR-T cells and combination therapies (immune checkpoint blockade therapy, radiation/chemotherapy, lytic virus therapy, and BKT inhibitors) are key to improving the efficacy of CAR-T. Combining multi-target CAR-T could also address tumor heterogeneity. In addition, more tumor-specific targets can be found by single-cell sequencing, thus avoiding off-target effects due to tumor heterogeneity.

The two primary toxicities in early animal studies were CRS and immune effector cell-associated neurotoxicity syndrome (ICANS) [115]. Among CAR-T cell therapies, CRS is one of the most common and serious toxicities in CAR-T cell therapy [116,117]. During CAR-T treatment, immune cells are stimulated with the subsequent release of a large number of cytokines, resulting in CRS, a significant systemic inflammatory response. The clinical manifestations vary, and the mild symptoms include diarrhea, anorexia, anorexia with or without fatigue, myalgia, and other generalized physical symptoms. Multiple organ systems, hypotension, arrhythmia, dyspnea, coagulation issues, and other symptoms might be present in severe instances [115,118,119]. ICANS is another common complication of CAR-T therapy [120,121]. However, the pathogenesis of ICANS is still unclear [122]. The possible cause is the increase in inflammatory cytokines and activation of endothelial cells, increasing vascular permeability and disrupting BBB, resulting in many cytokines and chemokines entering the CNS, thus causing injury [123]. The common symptom of ICANS is toxic encephalopathy. The early symptoms of ICANS are tremors, confusion, and inattention. In severe cases, it may progress to total aphasia, incontinence, severe disturbance of consciousness and movement, and seizures [124].

GBM immunosuppression of TME has been viewed as one of the main obstacles to CAR-T cell therapy’s effectiveness [125]. Both tumor and non-tumor cells are present in GBM, which creates a hostile environment that compromises T cell survival and function. It is a useful tactic to increase the release of proinflammatory cytokines to increase the activity of tumor cells. Preclinical studies on GBM have shown that co-expression of IL-15 [126], IL-12 [64], IL-18 [127], IL-23 [128], and IFNα2 [103] enhances the effect of CAR-T. However, excessive cytokine secretion can lead to CAR-T toxicity: CRS and ICANS. More recently, it has been suggested that directly targeting immunosuppressive cells in the TME, such as M2-type macrophages, is also a way to combat the TME [129]. In addition, enhancing NK cells and M1-type macrophages can improve the lethality of B7-H3 CAR-T cells through metabolic reprogramming of TME [86]. Modifying the physical structure and biochemical characteristics of the TME is also one of the effective strategies to improve the biological activity of CAR-T. The development and spread of solid tumors are aided by cancer-associated fibroblasts (CAFs), which stimulate angiogenesis. In 2015, Schuberth et al. developed CAR-T cells that specifically target the fibroblast-activating protein (FAP) to kill mesothelioma cells and inflammatory fibroblasts in vitro. These CAR-T cells successfully prevented the spread of FAP-positive human mesothelioma cells in the peritoneal cavity of mice and significantly increased the lifespan of mice [130]. Recently, Lee et al. developed radioactively labeled FAP inhibitors (FAPI) to improve FAP-CAR-T. The therapeutic impact of FAP-targeted CAR-T was dynamically detected using an in vitro non-invasive imaging approach [131]. Although these efforts have not yet been applied to GBM, they have provided novel ideas for researchers.

The off-target effect is also challenging in CAR-T cell therapy for treating GBM. Although many TAAs have been identified, such antigens are also expressed in normal cells to varying degrees, causing CAR-T cells to kill tumor cells while also damaging normal tissue cells. Therefore, finding suitable antigens as targets remains a key task for clinical practitioners. In addition to searching for highly specific antigens, the researchers utilized environmental factors to improve the CAR-T cell’s ability to recognize tumor cells specifically. It was possible to reduce the damage to normal cells expressing small amounts of target antigen while destroying tumor tissue using ultrasound to control CAR protein expression [114] (Figure 4).

## 5. Conclusions

CAR-T cell therapy is an exciting therapeutic option for solid malignancies. CAR-T cells have been demonstrated to be unsuccessful as monotherapy in clinical studies against numerous solid tumors due to immunological escape. It is anticipated that immunosuppressive TME could be overcome by combining techniques of radiation, chemotherapy, and other immunotherapies. Solid tumor inhibitors such as tumor heterogeneity, specific antigen deficiency, and immunosuppressive TME pose limitations and challenges to developing CAR-T cell immunotherapy. Due to the pathological characteristics of GBM, it is difficult for a single therapy to prevent the progression of the disease; therefore, a combination of multiple therapeutic methods is required to treat GBM. Current research focuses on finding TSAs, alleviating the off-target effects of CAR-T cells, improving the ability of CAR-T cells to combat the immunosuppressive TME, and reducing or eliminating the cytotoxicity of CAR-T cells.

## Figures and Tables

**Figure 1 cancers-15-02351-f001:**
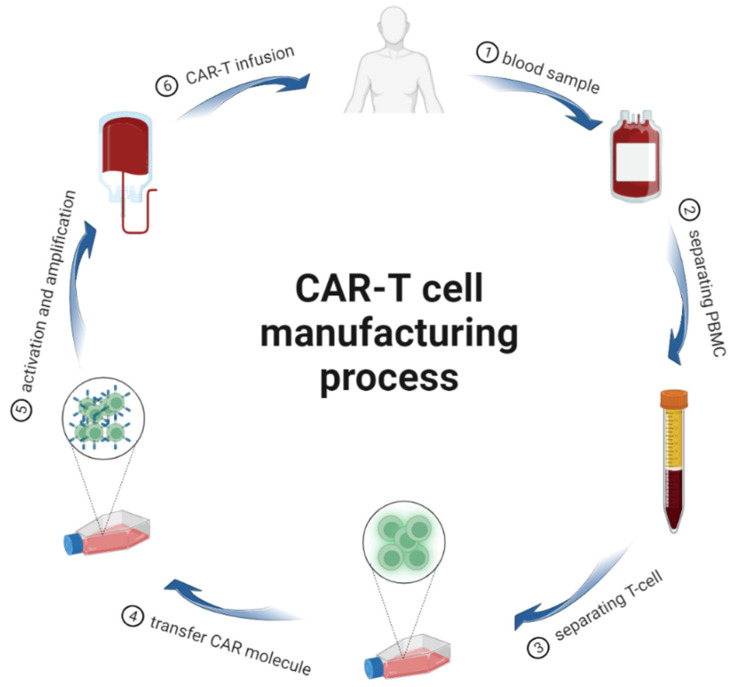
The CAR-T cell manufacturing process.

**Figure 2 cancers-15-02351-f002:**
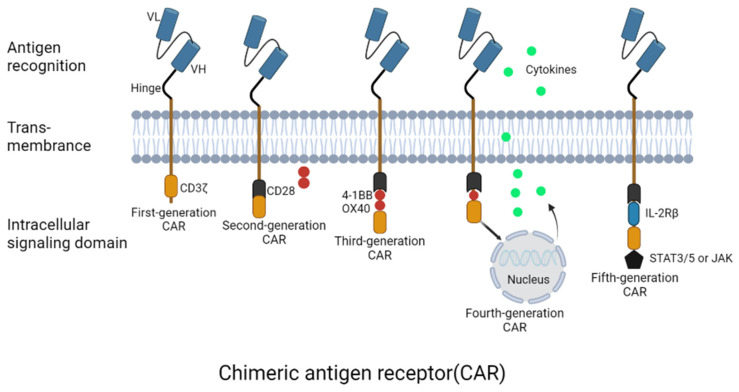
The design and evolution of different CAT-T cell generations. Three key structural domains are present in CAR-T cells: an external antigen-recognition domain, a transmembrane domain, and an intracellular signaling domain. In first-generation CARs, the T cell receptor’s (TCR) CD3 chain functions as an intracellular signaling domain alongside a transmembrane domain and an scFv targeting tumor-associated antigen (TAA) connected to a gap in an extracellular domain. By adding a co-stimulatory domain, such as CD28 or 4-1BB, the second-generation CARs increase the CAR-T cell activity. The third-generation CARs contain several co-stimulatory domains, such as OX-40 and 4-1BB. The fourth-generation CAR-T cells are designed to release engineered genes, such as cytokines, into tumor tissue when the CAR binds to a targeted antigen. The tumor immune microenvironment can be more effectively overcome. The fifth-generation CARs were designed to simultaneously activate the TCR, co-excitation domain CD28, and cytokine triple signaling, enhancing T cell proliferation, survival, and antitumor efficacy.

**Figure 3 cancers-15-02351-f003:**
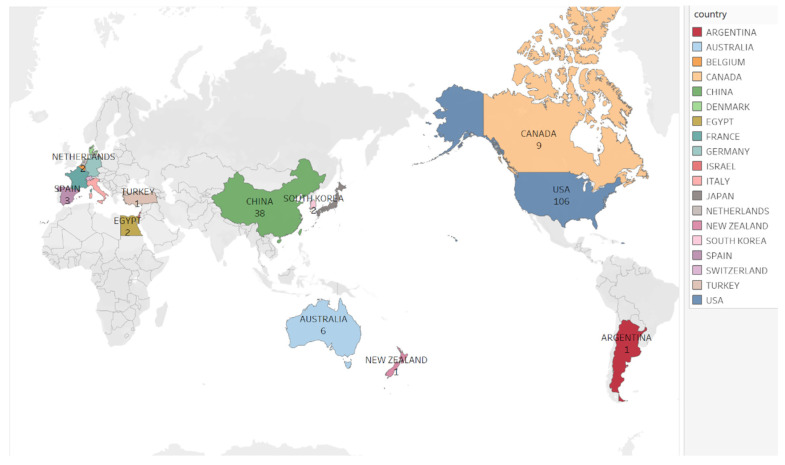
Total number of publications in the top 20 countries. USA 106, China 38, Germany 15, Canada 9, Japan 9, England 7, Australia 6, Italy 6, Switzerland 5, Spain 3, France 2, Egypt 2, South Korea 2, Denmark 2, Netherlands 2, Turkey 1, New Zealand 1, Israel 1, Belgium 1, Argentina 1.

**Figure 4 cancers-15-02351-f004:**
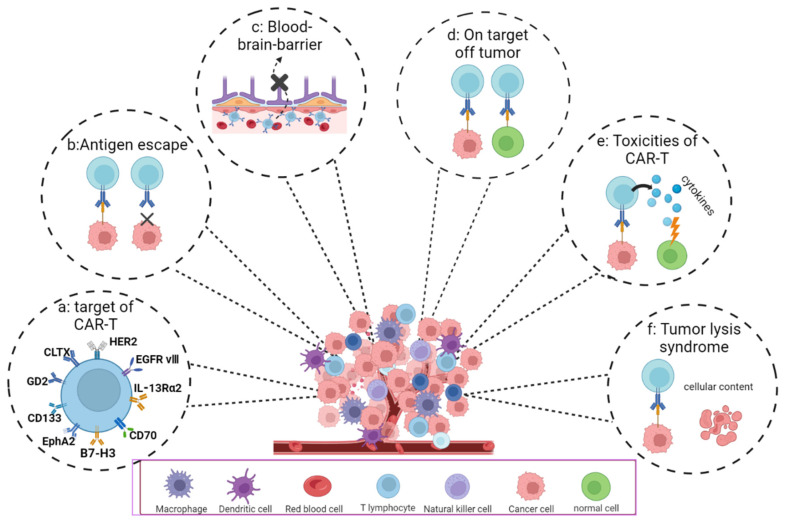
The difficulties and challenges of CAR-T cell therapy. (**a**) Gene targets of CAR-T. (**b**) Tumor heterogeneity and antigen escape. This is thought to be the underlying factor in drug resistance. (**c**) Several medications, including CAR-T, are prevented from entering the brain by the unique structure of the blood–brain barrier (BBB). (**d**) Off-target effect: CAR-T can recognize not only tumor cells, but also antigens shared with tumor cells on normal tissue cells to attack normal tissues and organs of the human body. (**e**) CAR-T cell-related toxicity, for example, cytokine release syndrome. (**f**) Tumor lysis syndrome (TLS). A large number of cancer cells that have been treated die in a short period of time and the cell contents are released into the bloodstream, leading to a variety of metabolic disorders.

**Table 1 cancers-15-02351-t001:** Published clinical trials of CAR-T for solid tumors and related targets.

Cancer Type	Target	Clinical Trial Phase	Main ID
Glioblastoma	EGFR vIII IL13Ra2 HER2 B7H3 CD70 GD2 MMP2 NKG2D	Phase I/Phase II Phase I Phase I Phase I/Phase II Phase I Phase I Not Applicable Phase I Phase I	NCT01454596 NCT04661384 NCT03500991 NCT04077866 NCT05366179 NCT05353530 ACTRN12622001514796 NCT05627323 NCT05131763
Gastrointestinal cancers	Claudin 18.2 CEA IM92	Phase I/Phase II Phase I/Phase II Early Phase I	NCT04404595 NCT05538195 NCT05275062
Lung cancer	GD2 B7H3 EGFR	Early Phase I Early Phase I Early Phase I	NCT05620342 NCT05341492 NCT05060796
Colorectal cance	CEA NKG2D	Phase I Phase I	NCT05240950 ChiCTR2100053018
Malignant pleural diseases	IL13Ra2	Phase I	NCT04119024
Renal cancer	CAIX CD70	Phase I Phase I	NCT04969354 NCT05420519
HCC	glypican-3	Phase I/Phase II	NCT05003895
Ovarian cancer	Mesothelin B7H3 TAG72	0 Phase I/Phase II Phase I	ChiCTR2100042320 NCT05211557 NCT05225363
Prostatic cancer	PSMA PSCA	Phase I Phase I	NCT05354375 NCT03873805
Metastatic melanoma	IL13Ra2	Phase I	NCT04119024
Breast Cancer	Mesothelin MCU1	Phase I Phase I/Phase II	NCT05623488 NCT02617134
Pancreatic cancer	IM92 B7H3 Claudin18.2	Early Phase I Phase I/Phase II Phase I/Phase II	NCT05275062 NCT05143151 NCT04404595

## Data Availability

Data openly available in a public repository.

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
