# Peer review of "Gene Targets of CAR-T Cell Therapy for Glioblastoma"

_cancers, 2023, doi:10.3390/cancers15082351_

Round 1
Reviewer 1 Report
This review attempts to provide an overview of the major explored targets for GBM and their therapeutic implications for CAR T therapy. While, it highlights the main targets of GBM and talks about the clinical trials involving these, it does not satisfactorily provide a thorough discussion on the challenges are of CAR T therapy in GBM and does not include the recent literature on some of the strategies to overcome these challenges. The language of the paper should be improved significantly.
Here are my specific comments:
The introduction provides a very basic overview of CAR T cells- It could be more concise. Avoid repetitive sentence structure. Add references to line 99 ( Fifth generation CAR)
Line 135-145: Include the other challenges of GBM in this discussion like Immunosupression, tumor heterogeneity, antigen loss etc.
In the TABLE- Specific which Phase of clinical trial if they are being conducted
Lines 142-152, including figure 3 is not relevant to the topic of discussion.
Elaborate on the major challenges for CART therapy in GBM, citing examples of clinical trials that have faced these challenges for each target. What are some of the recent advances in this field to specifically tackle these challenges- example: Sci Trans Med 2021 paper on SynNotch CAR talks about problems like specificity, persistence and heterogeneity- papers on overcoming Immunosuppression in TME. There a lot of recent literature on this that is not included in this review
Reviewer 2 Report
The review provides a clear description of different CAR-T cell therapy targets for glioblastoma. The introduction about glioma and CAR-T cell therapy is well-structured and lays down the foundation for the rest of the manuscript. The review presents interesting aspects of CAR-T therapy, through text and figures, while also discussing the challenges it faces in treating glioblastoma. Overall, the review is well written however some points require major attention.
1. The authors discuss only 1-2 studies pertaining to each target (for example: EGFRvIII, HER2, etc.) thereby missing out on many recent pre-clinical and completed clinical studies (Phase I or II) involving CAR-T therapy that have shown promise in treating glioblastoma. Therefore, it is suggested that the authors discuss recent studies related to the major target molecules (both pre-clinical and clinical) for potential readers.
2. The authors provide a table listing molecules targeted in different types of cancer besides glioblastoma. Do any of these molecules play a role in glioblastoma progression? If so, the authors are suggested to provide more information regarding these molecules and discuss them as potential targets for CAR-T therapy, as a separate section. Advillin antigens, mentioned on page 6, line 165 could also be described in the same section.
3. It is now well established that CD133 is not only present on stem cells like HSCs and CSCs but also on differentiated cells found in various tissues including pancreas, lungs and salivary glands, among many others. More importantly, contradicting reports describing the role of CD133 in GBM and other solid tumors have been published, pertaining to CAR-T therapy. It would be helpful to discuss the same since CD133 is implicated in various cancers.
4. Please provide an explanation for the paragraph on page 5 related to “national publications in the field” since its purpose is not clear with regards to the rest of the manuscript.
Minor comments:
1. The references in the text are not cited in a uniform manner. Most times they are at the end of the sentence however they also appear in the beginning of a sentence following the name of an author. This too is not uniform. Please compare line 177 and line 182 on page 6.
2. The status of clinical trial is missing for EGFRvIII in the Table.
3. In general, the text needs to be reviewed to correct small typing errors. For example: Page 9, line 314 states “CAR-t” therapy while the remaining text contains the term CAR-T therapy.
Reviewer 3 Report
The review by Wang et al., on targets of CART-T cell therapy for GBM appears to be conceptually all right. I have the following comments:
1. The ‘The’ in the title is not necessary. Or, maybe, say 'Gene targets of.....'
2. Provide a legend for Figure 1
3. What does the tick mark and the cross mark in the Table, column 4, mean?
4. Use different colors for the US and Argentina in Figure 3.
5. Change the title for section 3.1 to ‘Potential targets of CAR-T cell therapy for GBM’.
6. On line 167, last sentence should start as: In the following section…
7. Define EGFRvIII first – line170.
8. Line 183, provide full name for GCTO2.
9. Line 186: provide full name for C2173.
10. Lie 199: should be ‘poor prognosis’ and ‘poor survival’.
11. Line 226: clarify what is meant by – ‘At present, CAR molecules targeting IL-13alpha2 are mostly derived from mouse antibodies.’
12. Line 232: not sure what is meant by ‘This also avoids the adverse immune reaction of humans against mice’. There are other instances like this in the manuscript.
13. Also, mention if CAR-T approach has worked for ANY solid tumor. If it did, then mention what specific CAR-T approach was used, to what extent the clinical success achieved, etc.
Also, I would like the authors to speculate/discuss on the following scenario: Since GBM is heterogenous, say there are 20 different tumor cell types expressing 20 different surface antigens. However, the CAR-T approach is directed towards 5 cell types (5 antigens). What can we expect from such a treatment approach and what could be done in the future?
Reviewer 4 Report
The text by Wang et al. describes exhaustively the scientific and technological advances for the production of CAR-T with glioblastoma targets.
The authors initially describe CAR T production technology and haematological targets and then potential targets for solid tumors including glioblastoma. The authors then address specifically the potential antigenic targets present on the surface of cancer cells and the specific difficulties present in glioblastoma.
The description of the antigenic targets is very well articulated and also useful for the non-expert reader of the field.
A summary table with the main results obtained in the field of solid tumors including glioblastoma with CAR-T therapy would be useful.
Author Response
Dear reviewer,
Thank you for your decision and constructive comments on my manuscript.
Thank you very much for approving this article. Good luck with your work.
Sincerely
Chaoqun Wang.